# Life-time experience of violence among women and girls living with disability in Nepal

Padam Simkhada[1], Sapana Basnet[2], Shaurabh Sharma[3], Edwin van Teijlingen[4], Sharada Prasad Wasti[5], Tikadevi Dahal[6], Joshua Okyere[1,7]*, Ram Chandra Silwal[8], Manita Pyakurel[8]

1 School of Human & Health Sciences, University of Huddersfield, Huddersfield, United Kingdom, 2 Sightsavers International, West Sussex, United Kingdom, 3 Humanity & Inclusion – Handicap International, Kathmandu, Nepal, 4 Centre for Midwifery & Women's Health, Bournemouth University, Poole, United Kingdom, 5 School of Human Sciences, University of Greenwich, London, United Kingdom, 6 Nepal Disabled Women Association (NDWA), Kathmandu, Nepal, 7 Department of Population and Health, University of Cape Coast, Cape Coast, Ghana, 8 Green Tara Nepal, Kathmandu, Nepal

* joshuaokyere54@gmail.com/joshua.okyere@hud.ac.uk

## Abstract

### Background

Violence against women and girls with disabilities remains a serious, yet underexplored, global concern. This paper aims to investigate the prevalence and factors associated with life-time experience of violence within this vulnerable population in Nepal.

### Methods

This cross-sectional study was conducted in 28 municipalities representing all seven provinces as well as all three ecological regions of Nepal. A total of 1,294 women and girls with disability aged 15–59 years participated in this study. Data were collected in the period August to October 2021. This was done by trained enumerators using the KoBo application on smartphones or tablets. Both written and oral informed consent was sought from all participants. Each participant was assured of utmost confidentiality and privacy. Cross-tabulations were performed in STATA 18 to determine the distribution of the prevalence of violence. Also, bivariable and multivariable logistic regression models were fitted to establish association between the participants' characteristics and odds of experiencing violence.

### Results

Overall, 457 (35.32%) women living with disabilities had ever experienced violence at a point in their lifetime. Psychological/emotional violence was the most prevalent violence (74.40%) followed by physical violence (31.07%) and denial of services (28.67%). Age was positively associated with the likelihood of experiencing violence.

Data availability statement: The anonymized data is freely available at: https://doi.org/10.6084/m9.figshare.29266856.v2

Funding: The United Nations Women Trust Fund. This funding supported the study design, data collection and analysis. PS, SB, SS, and EVT secured the funding.

Competing interests: The authors have declared that no competing interests exist.

Abbreviations AOR: Adjusted Odds Ratio; PWDs: Persons with Disabilities; CI: Confidence Interval; NDWA: Nepal Disabled Women Association; NHRC: Nepal Health Research Council; WHO: World Health Organization.

Women belonging to the Brahman/Chhetri ethnic group had reduced odds of violence [AOR = 0.56; 95%CI: 0.37–0.85] compared to Hill Dalits. Divorced or separated women showed a markedly higher likelihood of experiencing violence [AOR = 6.69; 95%CI: 2.31–19.40] compared to currently married women. Participants who had not witnessed violence against other women exhibited significantly higher odds of experiencing violence [AOR = 1.86; 95%CI: 1.20–2.89]. Women living in the Koshi province [AOR = 4.04; 95%CI: 2.54–6.42], Madhesh province [AOR = 2.16; 95%CI: 1.15–4.08] and Bagmati province [AOR = 2.21; 95%CI: 1.41–3.46] reported significantly higher odds of experiencing violence compared to those in Karnali.

## Conclusion

The study concludes that age, ethnicity, marital status, and provincial residence are significant predictors of violence among women living with disability in Nepal. Interventions aimed at addressing violence against women living with disability in Nepal must prioritize older women and those who were previously married. Also, priority must be given to Koshi, Madhesh and Bagmati provinces where the prevalence and risk of experiencing violence is highest.

## Introduction

Violence against women and girls is a serious global health problem as highlighted by the World Health Organization [1]. Globally nearly 1.3 billion, or one in five people live with disability (PWDs), and most (80%) reside in the global south, and it is estimated that some three-quarters are female [2]. In Nepal, 2.2 percent of the total population live with at least one type of disability [3]; half of this population living with disabilities is female. To bring the issue of disability into lime light, the Government of Nepal in 2017 passed the Act on the Rights of Persons with Disabilities which categorizes disabilities into ten types based on the impairment (i.e., physical, vision related, hearing, deaf-blind, mental/psychological, intellectual disability, hemophilia, autism, and multiple disabilities), and also into four types based on the degree of severity [4].

PWDs are often one of the most disadvantaged populations in society [1,5,6]. In the general population of women, the violence against women in Nepal was exacerbated by the COVID-19 pandemic [7]. However, a recent review identified several important gaps in research around violence, such as lack of sex and disability disaggregation [8]. Violence against women and girls living with disabilities has adverse effects on their wellbeing, leading to a higher risk of severe distress, anxiety, and depression [9], and poorer self-reported health outcomes [10]. As such, it is imperative to understand the factors that exacerbates women with disabilities' risk of becoming victims of violence.

The 2021 Nepal Census records 654,782 PWDs in the country [10], and nearly half (45.8%) of them are females. In Nepal, the situation is exacerbated due to the caste system, in which individuals are stratified into disparate positions from birth by social and power structures [11,12]. A review by Thapa and colleagues

[12] highlighted that "Caste-based inequity impacts upon all aspects of an individual's well-being including violence and everyday life risks." Low-caste women and girls living with disabilities are more likely to experience gender-based violence (GBV) and discrimination [11]. To combat the problem Nepal established a Gender-Based Violence and Gender Equality Funds and adopted a gender-equality-and-development frame to facilitate the elimination of all forms of violence, including violence against PWDs [13,14]. There was also the implementation of the One-Stop Crisis Management Centers (OCMC) to provide a sanctuary for victims of violence [15].

Despite the implemented policies, studies conducted in Nepal have revealed a diverse prevalence of lifetime experiences of violence against women and girls with disabilities. For instance, Puri et al. [11] identified an overall lifetime prevalence of 57.7 percent across three districts, while Gupta et al. [16] reported 25.3 percent violence among married women in three districts of the Terai region. However, a significant drawback of these studies lies in their limited geographical focus, which does not offer a comprehensive understanding of the national level. Consequently, there is a pressing need for research that encompasses all provinces in Nepal, to provide a nationally representative perspective on this critical issue. This paper aims to investigate the prevalence and factors associated with life-time experience of violence within this vulnerable population in Nepal.

## Methods

### Study population and design

The study follows a quantitative research approach and specifically adopts a cross-sectional design. We conducted quantitative surveys in 14 districts representing all seven of Nepal's provinces (Koshi, Madhesh, Bagmati, Gandaki, Lumbini, Karnali, and Sudurpaschim). The study spanned districts in the three ecological areas. In the mountains area, we worked in Jumla, the districts in the Hills ecological area were: Dhankuta, Kavre, Kathmandu, Gorkha, Kaski, Argakhachi, Surkhet, and Achham. For the Terai ecological area, the following districts were covered: Morang, Siraha, Dhanusha, Dang, and Kanchanpur. A total of 1,294 women and girls with disability aged 15–59 years participated in this study. However, women and girls with severe intellectual disabilities were excluded as the research team did not have the capacity (i.e., trained interviewers in mental health) to survey women and girls with specialized needs.

### Sampling procedure

A multistage sampling technique [17] was applied, first, all districts were stratified based on (a) the three geographical regions and (b) seven provinces, next 14 districts were selected from these strata using a lottery method. In the second stage, 28 sites (16 municipalities, 9 rural municipalities, and 3 metropolitan cities) were chosen. Finally, a list of women and girls with disability for the selected municipalities was prepared with the help of members from the women and children development section of the municipality office. A representative from the Nepal Disabled Women Association (NDWA) and 18 female data enumerators collaboratively identified disabled women and girls' households for each area. As a result, 45–46 girls and women with disabilities were recruited from each rural municipality, urban municipality and metropolitan city. Participants had to have been residents of the study areas for at least 12 months preceding the survey.

It must be noted that this study is a baseline survey of a larger project in Nepal exploring multiple social and health dimensions of women and girls, including those living with disabilities. The sample size was informed by Puri et al.'s study [11] that found a 57.7% violence prevalence among women with disabilities. In our project, we estimated that there would be a 5% reduction during the project interventions. As such, the following parameters were considered in the sample size estimation: significance level = 0.05; power = 0.8; p1 = 0.577; p2 = 0.527 (5% decreased at end line); n2/n1(ratio) = 1; current mean cluster size = 4; intra-cluster correlation coefficient = 0.2; maximum expected cluster size = 5; minimum expected cluster size = 3; design effect: unadjusted = 1.6; and design effect: adjusted = 1.6125. This resulted in an adjusted sample size of 1,294.

## Study variables

Life-time experience of violence was the outcome variable. Lifetime is this context means that we are not restricted to the past 12 months or 1 month preceding the survey. Rather, we were interested in whether women living with disability had ever experienced any kind of violence at any point in their lifetime. This was derived from the question, "Have you ever experienced any form of violence?" The response categories were 'yes' and 'no'. Responding yes meant that the individual had ever experienced violence in their lifetime. Additionally, the specific violence type (i.e., physical violence, sexual violence, forced/child marriage, psychosocial abuse, and spousal violence) that has ever been experienced was also covered.

Based on the literature reviewed [8,11,16], nine potential explanatory variables we selected, including age (i.e., 15–19; 20–24; 25–29; 30–34; 35–39; and 40 years and over), ethnicity (i.e., Hill Dalit, Terai Dalit, Hill Janajati, Terai Janajati, Brahman/Chhetri), religion (i.e., Hindu, Buddhist, Muslim, Kirat and Christian), ever attended school, marital status (i.e., currently married, never married, divorced/separated, and widowed), age at first marriage (i.e., before 20 years, at 20 year or older), whether the person received disability allowance, knew policies and laws that support victims of violence, and ever witnessed violence against women. Disabilities were assessed across six functional domains: seeing, hearing, walking/climbing steps, remembering/concentrating, self-care, and communication. For each domain, participants were asked about the level of difficulty experienced, with responses categorized as: no difficulty, some difficulty, a lot of difficulty, or cannot do at all. Following the Washington Group recommendations, a binary variable was created for each domain, coded as 1 if the respondent reported "a lot of difficulty" or "cannot do at all," and 0 otherwise. Subsequently, a composite variable was generated to identify individuals with two or more disabilities by summing the six domain-specific binary variables and recoding the total as 1 if the individual had disabilities in two or more domains, and 0 otherwise.

## Data collection procedures

Study questionnaires were prepared and adopted from the Nepal Demographic and Health Survey, 2016 (NDHS, 2017), and Nepal Multiple Indicator Cluster Surveys 2014 (CBS, 2015) whose tools had already been used in Nepal. A pilot study [18] examined the adequacy of the questions, clarity/wording of questions, adequacy of possible responses (pre-coded), sequence/flow, and appropriate administration techniques. Based on the feedback from the pre-testing, all tools were finalized and used for the final data collection. Survey data were collected by trained enumerators using the KoBo application [19] on smartphones or tablets. Every day, data were sent to a web server by the enumerators and saved by the researcher in the period from August 30th to October 29th, 2021.

## Statistical analyses

The data keyed into KoBo, the computer-assisted personal interviewer, was extracted from the server and exported to STATA version 18 (StataCorp, College Station, TX, USA) for data management and analyses. Data were encoded to ensure that all numeric data were in their right format. Following that, a descriptive analysis was performed to determine the distribution of the sample and variables of interest. Pearson's chi-square test was then computed to determine whether there were significant differences in the estimated prevalence of life-time experience of violence across explanatory variables. As the outcome variable is binary in nature, binary logistic regression was fitted. First, the bivariable analysis was computed to ascertain the association between each explanatory variable and the likelihood of experiencing violence. The results were presented in odds ratio at a 95% confidence interval. Following that, a multivariable logistic regression model was fitted to adjust for the effect of all explanatory variables. This was represented by an adjusted odds ratio at a 95% confidence interval.

In the multivariable analysis, we employed a backward stepwise logistic regression approach to build the final adjusted model. Initially, all variables with theoretical relevance and those significant at p < 0.20 in the bivariate analysis were

included in the full model. Variables were then sequentially removed based on the likelihood ratio test (LRT) and Akaike Information Criterion (AIC) values, ensuring that removal did not meaningfully alter the effect estimates of key predictors or degrade the model's fit. Variables with p-values above 0.05 that did not contribute significantly to model fit and lacked theoretical justification were excluded. Specifically, religion, education, age at first marriage, disability allowance receipt, and knowledge on policies and laws supporting victims were excluded in the final model due to lack of statistical significance and negligible impact on the effect estimates of retained variables.

### Ethical considerations

Ethical approval was provided by the Nepal Health Research Council (NHRC) (registration number 339/2020P). Both written and oral informed consent was sought from all participants. Each participant was assured of utmost confidentiality and privacy. We assured them that all data collected will be anonymised to protect their identities.

## Results

### Characteristics of study participants

Table 1 shows the sample distribution. Most participants were aged 40 years and above (27.43%). In terms of ethnicity, the majority are Brahman/Chhetri (42.50%). Most participants identify as Hindu (88.25%), had not received any formal education (51.78%), and were never married (50.39%). However, a majority reported being married before the age of 20 years (82.61%). Most participants received a disability allowance (54.33%) and had ever witnessed violence against women (91.04%). Provincial distribution shows the highest representation from Gandaki Province (15.07%).

### Life-time prevalence of violence among women and girls living with disabilities

Overall, 457 (35.32%) women living with disabilities had ever experienced violence at a point in their lifetime (Table 1). Significantly high prevalence of violence was found among those age 30–34 years (43.56%), Terai Dalit participants (49.18%), and the Kirat group (83.33%). Divorced or separated individuals reported the highest prevalence of violence (82.86%). Persons who married at age 20 or older (53.49%) and those who lacked knowledge of policies on violence (36.39%) reported a high prevalence of violence. Interestingly, those who have not witnessed violence against women reported a substantially higher prevalence of personal violence (60.34%) compared to those who had. Provincially, participants from Koshi had the highest prevalence of violence (67.57%).

The prevalence of violence was generally higher among women with disabilities compared to those without, though the extent varied across disability types. Specifically, 42.08% of women with a visual disability reported experiencing violence, compared to those without a visual disability (33.62%). Similarly, 37.65% of women with a physical disability experienced violence, compared to those without such disability (33.33%). However, for self-care disability, the prevalence of violence was similar between women with the disability (33.92%) and those without (35.95%). In contrast, lower prevalence of violence was observed among women with hearing (23.66% vs. 36.22%), cognitive (24.46% vs. 36.62%), and communication disabilities (19.05% vs. 37.40%) compared to their counterparts without these disabilities.

### Types of violence experienced by women and girls living with disability

The results show that psychological/emotional violence was the most prevalent violence (74.40%) followed by physical violence (31.07%) and denial of services (28.67%). Sexual violence (2.19%) and forced/child marriage (2.63%) were the least prevalent type of violence among women living with disabilities (Table 2). Women in Karnali province (51.79%), Terai Dalit women (40.00%), those aged 30–34 years (38.03%), and those who professed Buddhism (39.29%) reported a higher prevalence of physical violence. Hill Janajati women reported a higher prevalence of psychological (78.24%), spousal (20.69%), and sexual violence (4.21%). Interestingly, participants with formal education reported high prevalence of all the types of violence.

**Table 1. Distribution of sample and experience of life-time prevalence of violence.**

| Characteristics | Sample n (%) | Proportion ever experienced violence N (%) | p-value |
|---|---|---|---|
| **Overall** | **1294 (100)** | **457 (35.32)** | |
| **Age** | | | **<0.001** |
| 15-19 years | 184 (14.22) | 39 (21.20) | |
| 20-24 years | 212 (16.38) | 75 (35.38) | |
| 25-29 years | 242 (18.70) | 95 (39.26) | |
| 30-34 years | 163 (12.60) | 71 (43.56) | |
| 35-39 years | 138 (10.66) | 56 (40.58) | |
| 40 years and above | 355 (27.43) | 121 (34.08) | |
| **Ethnicity** | | | **<0.001** |
| Hill Dalit | 190 (14.68) | 58 (30.53) | |
| Terai Dalit | 61 (4.71) | 30 (49.18) | |
| Hill Janajati | 224 (17.31) | 95 (42.41) | |
| Terai Janajati | 269 (20.79) | 114 (42.38) | |
| Brahman/Chhetri | 550 (42.50) | 160 (29.09) | |
| **Religion** | | | **<0.001** |
| Hindu | 1142 (88.25) | 374 (32.75) | |
| Buddhist | 47 (3.63) | 28 (59.57) | |
| Muslim | 33 (2.55) | 18 (54.55) | |
| Kirat | 12 (0.93) | 10 (83.33) | |
| Christian | 60 (4.64) | 27 (45.00) | |
| **Education** | | | 0.782 |
| No | 670 (51.78) | 239 (35.67) | |
| Yes | 624 (48.22) | 218 (34.94) | |
| **Marital status** | | | **<0.001** |
| Currently married | 550 (42.50) | 202 (36.73) | |
| Never married | 652 (50.39) | 206 (31.60) | |
| Divorced/separated | 35 (2.70) | 29 (82.86) | |
| Widowed | 57 (4.40) | 20 (35.09) | |
| **Age at first marriage** | | | **0.002** |
| Never married | 652 (50.39) | 206 (31.60) | |
| Before 20 years | 559 (46.29) | 228 (38.06) | |
| At 20 year or older | 43 (3.32) | 23 (53.49) | |
| **Receive disability allowance** | | | 0.308 |
| No | 591 (45.67) | 200 (33.84) | |
| Yes | 703 (54.33) | 257 (36.56) | |
| **Knowledge on policies and laws that support victims of violence** | | | 0.115 |
| Yes | 269 (20.79) | 84 (31.23) | |
| No | 1025 (79.21) | 373 (36.39) | |
| **Ever witnessed violence against women** | | | **<0.001** |
| Yes | 1178 (91.04) | 387 (32.85) | |
| No | 116 (8.96) | 70 (60.34) | |
| **Visual disability** | | | **0.011** |
| No | 1035 (79.98) | 348 (33.62) | |
| Yes | 259 (20.02) | 109 (42.08) | |

*(Continued)*

**Table 1.** (Continued)

| Characteristics | Sample n (%) | Proportion ever experienced violence N (%) | p-value |
|---|---|---|---|
| **Hearing disability** | | | **0.015** |
| No | 1201 (92.81) | 435 (36.22) | |
| Yes | 93 (7.19) | 22 (23.66) | |
| **Physical disability** | | | 0.106 |
| No | 699 (54.02) | 233 (33.33) | |
| Yes | 595 (45.98) | 224 (37.65) | |
| **Cognitive disability** | | | **0.005** |
| No | 1155 (89.26) | 423 (36.62) | |
| Yes | 139 (10.74) | 34 (24.46) | |
| **Self-care disability** | | | 0.480 |
| No | 893 (69.01) | 321 (35.95) | |
| Yes | 401 (30.99) | 136 (33.92) | |
| **Communication disability** | | | **<0.001** |
| No | 1147 (88.64) | 429 (37.40) | |
| Yes | 147 (11.36) | 28 (19.05) | |
| **Two or more disabilities** | | | 0.967 |
| No | 857 (66.23) | 303 (35.36) | |
| Yes | 437 (33.77) | 154 (35.24) | |
| **Provinces** | | | **<0.001** |
| Koshi Province | 185 (14.30) | 125 (67.57) | |
| Madhesh Province | 182 (14.06) | 96 (52.75) | |
| Bagmati Province | 187 (14.45) | 90 (48.13) | |
| Gandaki Province | 195 (15.07) | 51 (26.15) | |
| Lumbini Province | 187 (14.45) | 26 (13.90) | |
| Karnali Province | 190 (14.68) | 56 (29.47) | |
| Sudurpaschim Province | 168 (12.98) | 13 (7.74) | |

### Factors associated with violence against women living with disabilities

Table 3 shows the results for the bivariable and multivariable logistic regression models. In the adjusted model, age was positively associated with the likelihood of experiencing violence, with women aged 25–29 years [AOR = 2.17; 95%CI: 1.29–3.65], 30–34 years [AOR = 2.66; 95%CI: 1.51–4.68], and 35–39 years [AOR = 2.16; 95%CI: 1.20–3.90] having significantly higher odds compared to those aged 15–19 years. Women belonging to the Brahman/Chhetri ethnic group had reduced odds of violence [AOR = 0.56; 95%CI: 0.37–0.85] compared to Hill Dalits. Divorced or separated women showed a markedly higher likelihood of experiencing violence [AOR = 6.69; 95%CI: 2.31–19.40] compared to currently married women. Participants who had not witnessed violence against other women exhibited significantly higher odds of experiencing violence [AOR = 1.86; 95%CI: 1.20–2.89]. Women living in the Koshi province [AOR = 4.04; 95%CI: 2.54–6.42], Madhesh province [AOR = 2.16; 95%CI: 1.15–4.08] and Bagmati province [AOR = 2.21; 95%CI: 1.41–3.46] reported significantly higher odds of experiencing violence compared to those in Karnali.

## Discussion

This study investigated the lifetime prevalence of violence against women and girls living with disabilities in Nepal. The results show that 35.32% of the participants had ever experienced violence in their lifetime. While the estimated

**Table 2. Distribution of the prevalence of the types of violence among women living with disability.**

| Characteristics | Sexual violence n (%) | Physical violence n (%) | Force/child marriage n (%) | Denial of services n (%) | Spousal violence n (%) | Psychological violence n (%) |
|---|---|---|---|---|---|---|
| **Overall** | **10 (2.19)** | **142 (31.07)** | **12 (2.63)** | **131 (28.67)** | **61 (13.35)** | **340 (74.40)** |
| **Age** | | | | | | |
| 15-19 years | 3 (7.69) | 12 (30.77) | 1 (2.56) | 10 (25.64) | 1 (2.56) | 31 (79.49) |
| 20-24 years | 2 (2.67) | 19 (25.33) | 1 (1.33) | 18 (24.00) | 4 (5.33) | 57 (76.00) |
| 25-29 years | 2 (2.11) | 31 (32.63) | 3 (3.16) | 30 (31.58) | 14 (14.74) | 67 (70.53) |
| 30-34 years | 2 (2.82) | 27 (38.03) | 1 (1.41) | 22 (30.99) | 11 (15.49) | 53 (74.65) |
| 35-39 years | 0 (0.00) | 15 (26.79) | 0 (0.00) | 16 (28.57) | 8 (14.29) | 46 (82.14) |
| 40 years and above | 1 (0.83) | 38 (31.40) | 6 (4.96) | 35 (28.93) | 23 (19.01) | 86 (71.07) |
| **Ethnicity** | | | | | | |
| Hill Dalit | 2 (3.45) | 20 (34.48) | 1 (1.72) | 14 (24.14) | 12 (20.69) | 43 (74.14) |
| Terai Dalit | 0 (0.00) | 12 (40.00) | 0 (0.00) | 15 (50.00) | 5 (16.67) | 21 (70.00) |
| Hill Janajati | 4 (4.21) | 33 (34.74) | 1 (1.05) | 26 (27.37) | 13 (13.68) | 71 (74.74) |
| Terai Janajati | 1 (0.88) | 34 (29.82) | 2 (1.75) | 34 (29.82) | 15 (13.16) | 80 (70.18) |
| Brahman/Chhetri | 3 (1.88) | 43 (26.88) | 8 (5.00) | 42 (26.25) | 16 (10.00) | 125 (78.12) |
| **Religion** | | | | | | |
| Hindu | 7 (1.87) | 114 (30.48) | 10 (2.67) | 115 (30.75) | 48 (12.83) | 277 (74.06) |
| Buddhist | 1 (3.57) | 11 (39.29) | 0 (0.00) | 5 (17.86) | 7 (25.00) | 23 (82.14) |
| Muslim | 1 (5.56) | 6 (33.33) | 2 (11.11) | 3 (16.67) | 3 (22.22) | 10 (55.56) |
| Kirat | 0 (0.00) | 2 (20.00) | 0 (0.00) | 5 (50.00) | 0 (0.00) | 10 (100.0) |
| Christian | 1 (3.70) | 9 (33.33) | 0 (0.00) | 3 (11.11) | 2 (7.41) | 20 (74.07) |
| **Education** | | | | | | |
| No | 4 (1.67) | 65 (27.20) | 4 (1.67) | 58 (24.27) | 27 (11.30) | 187 (78.24) |
| Yes | 6 (2.75) | 77 (35.32) | 8 (3.67) | 73 (33.49) | 34 (15.60) | 153 (70.18) |
| **Provinces** | | | | | | |
| Koshi Province | 2 (1.60) | 28 (22.40) | 2 (1.60) | 54 (43.20) | 13 (10.40) | 117 (93.60) |
| Madhesh Province | 0 (0.00) | 31 (32.29) | 0 (0.00) | 45 (46.88) | 17 (17.71) | 72 (75.00) |
| Bagmati Province | 3 (3.33) | 23 (25.56) | 3 (3.33) | 15 (16.67) | 4 (4.44) | 62 (68.89) |
| Gandaki Province | 0 (0.00) | 16 (31.37) | 0 (0.00) | 7 (13.73) | 10 (19.61) | 35 (68.63) |
| Lumbini Province | 5 (19.23) | 11 (42.31) | 3 (11.54) | 2 (7.69) | 2 (7.69) | 4 (15.38) |
| Karnali Province | 0 (0.00) | 29 (51.79) | 4 (7.14) | 7 (12.50) | 9 (16.07) | 42 (75.00) |
| Sudurpaschim Province | 0 (0.00) | 4 (30.77) | 0 (0.00) | 1 (7.69) | 6 (46.15) | 8 (61.54) |

NB: This was a multi-select question answered by 457 participants who had every experienced any violence

prevalence of violence among women living with disabilities is lower than what has been previously reported in another Nepalese study (58%) [11], it resonates with the global statistics where one in three women experience violence [1]. The findings also align with a recent systematic review of 29 LMICs that showed that women with disabilities are twice as likely to experience violence compared to their counterparts without disabilities [20]. The results also resonate with that of developed countries like Australia, and the USA [21,22]. Thus, emphasizing the immediate necessity for focused interventions and assistance programs for girls with disabilities in Nepal.

Consistent with previous studies [1,23,24], the present study found psychological violence to be the most prevalent violence type, followed by physical violence. The low prevalence of sexual violence observed in this study could be explained

**Table 3. Factors associated with violence against women living with disabilities.**

| Variables | Unadjusted Model OR [95%CI] | Adjusted Model AOR [95%CI] |
|---|---|---|
| **Age** | | |
| 15-19 years | Ref. | Ref. |
| 20-24 years | **2.03 [1.29-3.19]**** | **1.78 [1.06-2.98]*** |
| 25-29 years | **2.40 [1.55-3.72]***** | **2.17 [1.29-3.65]**** |
| 30-34 years | **2.87 [1.79-4.59]***** | **2.66 [1.51-4.68]**** |
| 35-39 years | **2.54 [1.55-4.15]***** | **2.16 [1.20-3.90]*** |
| 40 years and above | **1.92 [1.27-2.91]**** | **1.88 [1.10-3.21]*** |
| **Ethnicity** | | |
| Hill Dalit | Ref. | Ref. |
| Terai Dalit | **2.20 [1.22-3.97]**** | 0.91 [0.40-2.06] |
| Hill Janajati | **1.68 [1.11-2.52]*** | 0.79 [0.49-1.27] |
| Terai Janajati | **1.67 [1.13-2.48]*** | 0.89 [0.51-1.56] |
| Brahman/Chhetri | 0.93 [0.65-1.34] | **0.56 [0.37-0.85]**** |
| **Religion** | | |
| Hindu | Ref. | – |
| Buddhist | **3.03 [1.67-5.49]***** | – |
| Muslim | **2.46 [1.23-4.94]*** | – |
| Kirat | **10.27 [2.24-47.12]**** | – |
| Christian | 1.68 [0.99-2.84] | – |
| **Education** | | |
| No | Ref. | – |
| Yes | 0.97 [0.77-1.22] | – |
| **Marital status** | | |
| Currently married | Ref. | Ref. |
| Never married | 0.79 [0.63-1.01] | 0.89 [0.66-1.21] |
| Divorced/separated | **8.33 [3.39-20.40]***** | **6.69 [2.31-19.40]***** |
| Widowed | 0.93 [0.53-1.65] | 1.51 [0.73-3.12] |
| **Age at first marriage** | | |
| Never married | Ref. | |
| Before 20 years | **0.75 [0.59-0.95]*** | – |
| At 20 year or older | **0.40 [0.21-0.75]**** | – |
| **Receive disability allowance** | | |
| No | Ref. | – |
| Yes | 1.13 [0.89-1.42] | – |
| **Knowledge on policies and laws that support victims of violence** | | |
| Yes | Ref. | – |
| No | 1.26 [0.94-1.68] | – |
| **Ever witnessed violence against women** | | |
| Yes | Ref. | Ref. |
| No | **3.11 [2.10-4.60]***** | **1.86 [1.20-2.89]**** |
| **Provinces** | | |
| Koshi Province | **4.98 [3.21-7.73]***** | **4.04 [2.54-6.42]***** |
| Madhesh Province | **2.67 [1.74-4.09]***** | **2.16 [1.15-4.08]*** |
| Bagmati Province | **2.22 [1.45-3.39]***** | **2.21 [1.41-3.46]**** |

*(Continued)*

**Table 3.** (Continued)

| Variables | Unadjusted Model OR [95%CI] | Adjusted Model AOR [95%CI] |
|---|---|---|
| Gandaki Province | 0.85 [0.54-1.32] | 0.84 [0.53-1.34] |
| Lumbini Province | **0.39 [0.23-0.65]**\*** | **0.34 [0.19-0.61]**\*** |
| Karnali Province | Ref. | Ref. |
| Sudurpaschim Province | **0.20 [0.10-0.38]**\*** | **0.18 [0.09-0.35]**\*** |

\*$p<0.05$, \*\*$p<0.01$, \*\*\*$p<0.001$; Ref: Reference category

**(-)** *Variables excluded after backward stepwise approach*

from the perspective that victims of such violence are highly stigmatized and thus, are unlikely to report their experience for fear of shame, blame, or ostracization from their families and communities [23,25]. Nonetheless, compared to other types of disabilities, sexual violence appears to be pervasive among those living with intellectual disabilities. Our observation aligns with previous literature that have found sexual violence to four times more prevalent among persons living with intellectual disabilities than any other group in the disability spectrum [26,27]. Perhaps, the reasoning challenges of women with intellectual disabilities connote a notion that they would be unable to report acts of sexual violence. As such, perpetrators of this violence take advantage of their cognitive impairments.

Women living with vocal or speech disabilities were more vulnerable to denial of services. This finding corroborates Stransky et al.'s study [28] that found that individuals with communication disabilities, including speech impairments, had a difficult time accessing healthcare services. It is plausible that this finding reflects the lack of inclusive service systems that accommodate alternative communication methods such as sign language and augmentative devices. It is also possible that members of the society may perceive individuals with vocal and speech-related disabilities as less capable of benefiting from certain services, leading to their exclusion from key services (e.g., sexual and reproductive health services, financial services, etc.). The elevated prevalence of forced or child marriage among the deaf-blind may be tied to societal biases perceiving them as burdens; hence, families lean towards such arrangements to reduce caregiving responsibilities.

As the ages of women with disability increases, the likelihood of them being exposed to violence increases. This is inconsistent with a previous study that found an inverse association between age and risk of violence victimization [29]. The difference between our findings and that of Brownridge [29] stems from the latter focusing on experience of violence in the last five years, whilst our study focused on life-time experience. And so, our finding reflects the cumulative nature of lifetime exposure, where the probability of ever experiencing violence increases with age due to a longer period of risk exposure. Therefore, this association should be interpreted cautiously, as it does not imply that older women are currently at higher risk of violence victimization compared to younger women but rather reflects accumulated experiences over the life course. The disaggregated data also revealed that sexual violence was more prevalent among young girls (15–19 years) than older women. Literature also shows that young girls living with disabilities in Nepal often lack access to sexual and reproductive health information and services. As such, they are less likely to be sexually empowered, and this exacerbates their vulnerabilities to sexual violence [30].

The results revealed that compared to currently married women, those who were previously married (i.e., divorced/separated) were significantly more likely to have experienced violence. This is inconsistent with previous studies [31,32] that found that married people are more exposed to violence. However, the result is corroborated by one study that shows a high prevalence of risk of violence among divorced women [33]. One plausible interpretation may be that divorced women are more likely to be subjected to ridicule, verbal abuse and in some cases, denied access to critical services as they are believed to bring bad luck [34].

Compared to Karnali Province, women with disabilities in the Koshi, Madhesh and Bagmati provinces were significantly more likely to experience violence. To the best of our knowledge, there are no published studies directly comparing this observation. However, a study by Kakchapati et al. [30] reported that women with disabilities in Karnali and Sudurpaschim

have greater access to sexual and reproductive health services. This access may indirectly contribute to reduced vulnerability to violence by exposing women to education on violence prevention, recognition of different forms of violence, and strategies to protect themselves in such circumstances.

In contrast to earlier studies that have found a significant inverse association between education and risk of violence among women with disabilities [35,36], there was no significant association in this study. Also, women with formal education had a higher prevalence of experiencing violence. Our finding is synonymous to Puri et al.'s study [11] that found no statistically significant association between education and risk of experiencing violence among persons living with disabilities in Nepal. One possible explanation is that formal education may increase visibility and social interactions, inadvertently exposing women with disabilities to a broader spectrum of societal discrimination and violence. Moreover, systemic factors such as stigmatization or lack of enforcement of disability rights may undermine the protective role of education.

## Implications for Policy and Practice

Evidence from this study underscores a need for the Nepalese government to prioritize the development and enforcement of comprehensive, disability-inclusive anti-violence legislation that addresses the unique risks identified in this study. For instance, targeted interventions are essential to protect women with intellectual disabilities from sexual violence, including community education programs to challenge stigma and enforce strict penalties against perpetrators who exploit cognitive impairments. Additionally, there is an urgent need to make service delivery systems more inclusive by incorporating alternative communication methods, such as sign language and augmentative communication devices, to reduce the exclusion experienced by women with vocal and speech-related disabilities. The provincial disparities highlight a need to assess the individual provinces' current interventions aimed at preventing violence against women living with disabilities. Specifically, Koshi could learn some best practices from the other provinces such as Sudurpaschim.

## Strengths and limitations

The focus of the study on PWDs is one of the strengths as it exposes a less explored area in Nepal. Also, the sample size is large enough to make extrapolations to the larger population of women and girls living with disability in Nepal. Nonetheless, the cross-sectional nature precludes us from making any sort of causal inferences. Also, as this was a quantitative study, the role of cultural beliefs and normative systems in influencing the experience of violence among this vulnerable population is not accounted for. Also, the study does not include males or people over 60. Consequently, the findings may not be generalizable to these extended populations. Additionally, the self-reported nature of the study and the fact that participants had to recall their experience of violence may have resulted in some recall bias and social desirability biases. Another limitation of this study lies in how life-time experience of violence was measured. There was no prompting in the questioning. As such, it is possible that some participants may not recognize some actions (e.g., denial of service) as violence. Therefore, it is possible that the prevalence may be underestimated in this study.

## Conclusion

The study concludes that age, ethnicity, marital status, and provincial residence are significant predictors of violence among women living with disability in Nepal. Interventions aimed at addressing violence against women living with disability in Nepal must prioritize older women and those who were previously married. Also, priority must be given to Koshi, Madhesh and Bagmati provinces where the prevalence and risk of experiencing violence is highest.

## Author contributions

**Conceptualization:** Padam Simkhada, Sapana Basnet, Shaurabh Sharma, Edwin van Teijlingen, Sharada Prasad Wasti, Ram Chandra Silwal.

**Data curation:** Joshua Okyere.

**Formal analysis:** Joshua Okyere.

**Funding acquisition:** Padam Simkhada, Sapana Basnet, Shaurabh Sharma, Edwin van Teijlingen, Ram Chandra Silwal.

**Investigation:** Padam Simkhada, Sapana Basnet, Shaurabh Sharma, Edwin van Teijlingen, Tikadevi Dahal, Manita Pyakurel, Ram Chandra Silwal.

**Methodology:** Padam Simkhada, Sapana Basnet, Shaurabh Sharma, Edwin van Teijlingen, Sharada Prasad Wasti, Tikadevi Dahal, Joshua Okyere, Manita Pyakurel, Ram Chandra Silwal.

**Project administration:** Padam Simkhada, Edwin van Teijlingen, Sharada Prasad Wasti.

**Software:** Joshua Okyere.

**Supervision:** Padam Simkhada, Edwin van Teijlingen.

**Validation:** Tikadevi Dahal.

**Writing – original draft:** Padam Simkhada, Sapana Basnet, Shaurabh Sharma, Edwin van Teijlingen, Sharada Prasad Wasti, Tikadevi Dahal, Joshua Okyere, Manita Pyakurel.

**Writing – review & editing:** Padam Simkhada, Sapana Basnet, Shaurabh Sharma, Edwin van Teijlingen, Sharada Prasad Wasti, Tikadevi Dahal, Joshua Okyere, Manita Pyakurel, Ram Chandra Silwal.

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
