## [Decision Letter · Decision Letter 0]

PLOS ONE

We look forward to receiving your revised manuscript.

Kind regards,

Shalik Ram Dhital, PhD

Academic Editor

PLOS ONE

2. In this instance it seems there may be acceptable restrictions in place that prevent the public sharing of your minimal data. However, in line with our goal of ensuring long-term data availability to all interested researchers, PLOS’ Data Policy states that authors cannot be the sole named individuals responsible for ensuring data access (http://journals.plos.org/plosone/s/data-availability#loc-acceptable-data-sharing-methods).

Reviewers' comments:

Reviewer's Responses to Questions

**Comments to the Author**

1. Is the manuscript technically sound, and do the data support the conclusions?

Reviewer #1: Yes

Reviewer #2: Yes

2. Has the statistical analysis been performed appropriately and rigorously?

Reviewer #1: Yes

Reviewer #2: Yes

3. Have the authors made all data underlying the findings in their manuscript fully available?

Reviewer #1: Yes

Reviewer #2: Yes

4. Is the manuscript presented in an intelligible fashion and written in standard English?

Reviewer #1: Yes

Reviewer #2: Yes

Reviewer #1: Manuscript Review Report

Manuscript Number: PONE-D-24-24275

Title: Life-time experience of violence among women and girls living with disability in Nepal: A cross-sectional study

Date of Review: 12th July 2024

General Comments:

• Relevance: The study is highly relevant to the present context, addressing a critical issue affecting a vulnerable population in Nepal.

• Statistical Analysis: The statistical analysis performed is appropriate for the study design and objectives.

• Language Quality and Presentation: The manuscript is generally satisfactory in terms of language quality and presentation.

1. Author Information:

• Educational qualifications of authors other than those with a Ph.D. are not provided.

o Recommendation: Include the educational qualifications of all authors to provide a complete overview of their expertise and contributions.

2. Abstract:

• The abstract states, “Descriptive and multivariate statistical analysis for association was performed to ascertain the effect of each explanatory variable.” It does not mention bivariate analysis.

o Recommendation: Clarify whether bivariate analysis was conducted and, if so, mention it in the abstract.

3. Keywords:

• The current keywords are not entirely appropriate.

o Recommendation: Use the following suggested keywords for better indexing:

Violence against Women

Violence against Girls

Women with Disabilities

Girls with Disabilities

Cross-sectional Survey

South Asia

Nepal

4. Methodology:

• The study design does not specify which districts, ecological regions, and provinces are included.

o Recommendation: Clearly state the districts, ecological regions, and provinces covered in the study to provide a detailed context of the study area.

• The method for determining the sample size is not explained.

o Recommendation: Describe the process used to determine the sample size, including any calculations or criteria used.

5. Dependent Variable:

• The lifetime experience of violence as an outcome variable is derived from a broad question, “Have you ever experienced any form of violence?” This is very general.

o Recommendation: Specify the types and forms of violence (physical, sexual, emotional) considered. If a standard tool for measuring lifetime violence experience exists, mention it or justify why a single question is sufficient.

• The operational definition of the dependent variable (lifetime violence) is unclear.

o Recommendation: Provide a clear operational definition of lifetime violence, specifying the types and forms included.

• The lifetime prevalence of violence may have recall bias.

o Recommendation: Acknowledge and discuss the potential for recall bias in the manuscript.

6. Results:

• The symbol for Chi-square is incorrect.

o Recommendation: Correct the symbol for Chi-square in the results section.

• The results are presented appropriately.

Overall Assessment:

• The manuscript addresses an important issue and uses appropriate statistical analysis. However, it requires revisions to improve clarity and detail in the methodology, keyword selection, and operational definitions.

Reviewer #2: I greatly appreciate the authors for addressing the issue of violence against women with disabilities—a topic that is often overlooked and given low priority in our context. This study provides valuable insights into the prevalence and nature of such violence. While the paper is well-organized and adheres to journal standards, I have some observations and suggestions for improvement to make it more comprehensive.

Abstract

• While the abstract mentions that half of women with disabilities have experienced violence, it would be more insightful to specify which age groups are most affected and the types of violence they face, such as sexual, physical, or others, rather than focusing solely on lower odds. Additionally, there is a lack of coherence between the abstract and recommendations. The abstract emphasizes lower odds, while the recommendations highlight a high prevalence of violence and propose policies incorporating intersectionality. This inconsistency could be addressed to improve clarity and alignment.

Background

• Referencing within the background section is inconsistent. For example, the sixth line references a United Nations report within the text, whereas other sections adopt the Vancouver style.

• Moreover, I recommend adding background information on the specific types of violence faced by women with disabilities—such as sexual, physical, or domestic violence, globally, regionally or if possible from Nepal to underline the significance of the study and its policy implications. Also, a brief discussion category of the Government of Nepal for disability, and the category that study has consider so that reader can understand why some of the disability is not consider or vice versa.

Methods

• The categorization of disabilities could be improved by aligning it with the Nepalese government's guidelines, which classify disabilities into ten types. The authors should describe this framework in both the background and methods sections to provide clarity. Including a table with operational definitions for the types of violence and disabilities would also enhance the paper's comprehensiveness.

Additionally, the term "Brahman" should be used consistently instead of "Braman" to align with the terminology in Nepal’s national population and housing census.

Results

• Figures are missing from the results section, which makes it difficult to provide specific feedback. However, I suggest using the term "vocal and speech-related disability" rather than "speech/voice disability" for standard terminology.

It is notable that violence is more prevalent among younger women with disabilities compared to older ones in this study which is commonly seen as well. Therefore the discussion should explore the types of violence younger women are more susceptible to, such as sexual violence or neglect (e.g., menstrual hygiene), as this specificity would provide valuable insights.

• While education status is not statistically significant in relation to violence, the fact that violence is slightly higher among educated women is surprising. A potential explanation could be their increased social engagement, which may expose them to stigma or bullying by peers and teachers. Discussing such non-significant in number yet intriguing findings would enrich the paper.

Discussion

• The discussion section is well presented. However, I recommend incorporating an analysis of the types of violence experienced by specific age groups and discuss it more and according to provinces too. This would enable Nepal to develop targeted, evidence-based policies that are contextualized to local realities.

Conclusion

• The conclusion feels overly generic. While the results highlight factors associated with violence, the conclusion should provide more specific recommendations. For example, critical issues like denial of services should be addressed explicitly, alongside suggestions for law reform. Concluding with a focus on the significance of further research and its potential impact would make the conclusion more impactful and comprehensive.

**Do you want your identity to be public for this peer review?** For information about this choice, including consent withdrawal, please see our Privacy Policy

Reviewer #1: **Yes: ** Dr. Hari Prasad Kaphle, Associate Professor (Public Health), Pokhara University

Reviewer #2: No

---

## [Author Response · Author response to Decision Letter 1]

7 Jan 2025

Response

Reviewer #1: Manuscript Review Report

Manuscript Number: PONE-D-24-24275

Title: Life-time experience of violence among women and girls living with disability in Nepal: A cross-sectional study

Date of Review: 12th July 2024

General Comments:

• Relevance: The study is highly relevant to the present context, addressing a critical issue affecting a vulnerable population in Nepal.

Response: Thank you for taking time to review our manuscript and provide appropriate feedback.

• Statistical Analysis: The statistical analysis performed is appropriate for the study design and objectives.

Response: Thank you.

• Language Quality and Presentation: The manuscript is generally satisfactory in terms of language quality and presentation.

Response: Thank you.

1. Author Information:

• Educational qualifications of authors other than those with a Ph.D. are not provided.

o Recommendation: Include the educational qualifications of all authors to provide a complete overview of their expertise and contributions.

Response: We have now provided the educational qualifications of all authors.

2. Abstract:

• The abstract states, “Descriptive and multivariate statistical analysis for association was performed to ascertain the effect of each explanatory variable.” It does not mention bivariate analysis.

o Recommendation: Clarify whether bivariate analysis was conducted and, if so, mention it in the abstract.

Response: We have now clarified this. It now reads: “Cross-tabulations and chi-square tests were performed in STATA 18. Also, bivariable and multivariable logistic regression models were fitted to establish associations.”

3. Keywords:

• The current keywords are not entirely appropriate.

o Recommendation: Use the following suggested keywords for better indexing:

Violence against Women

Violence against Girls

Women with Disabilities

Girls with Disabilities

Cross-sectional Survey

South Asia

Nepal

Response: We have now revised the keywords: “Violence Against Women; Girls with Disability; Cross-sectional Survey, South Asia; Nepal”

4. Methodology:

• The study design does not specify which districts, ecological regions, and provinces are included.

o Recommendation: Clearly state the districts, ecological regions, and provinces covered in the study to provide a detailed context of the study area.

Response: We have now provided this detail. It reads: “We conducted quantitative surveys in 14 districts representing all seven of Nepal’s provinces (Province 1, Province 2, Bagmati, Gandaki, Lumbini, Karnali, and Sudurpaschim). The study spanned districts in the three ecological areas. In the mountains area, we worked in Jumla. Also, the following districts were in the Hills ecological area; Dhankuta, Kavre, Kathmandu, Gorkha, Kaski, Argakhachi, Surkhet, and Achham. For the Terai ecological area, the following districts were covered: Morang, Siraha, Dhanusha, Dang, and Kanchanpur.”

• The method for determining the sample size is not explained.

o Recommendation: Describe the process used to determine the sample size, including any calculations or criteria used.

Response: Thank you for the comment. We have now provided this detail: “It must be noted that this study is part of a larger project in Nepal to reduce violence against women with disabilities. The sample size was informed by Puri et al. [11] study that found a 57.7% violence prevalence among women with disabilities. In our project, we estimated that were would be a 5% reduction during the project interventions. As such, the following parameters were considered in the sample size estimation: significance level=0.05; power=0.8; p1 = 0.577; p2 = 0.527 (5% decreased at end line); n2/n1(ratio) = 1; current mean cluster size = 4; intra-cluster correlation coefficient = 0.2; maximum expected cluster size = 5; minimum expected cluster size = 3; design effect: unadjusted = 1.6; and design effect: adjusted = 1.6125. This resulted in an adjusted sample size of 1,294.”

5. Dependent Variable:

• The lifetime experience of violence as an outcome variable is derived from a broad question, “Have you ever experienced any form of violence?” This is very general.

o Recommendation: Specify the types and forms of violence (physical, sexual, emotional) considered. If a standard tool for measuring lifetime violence experience exists, mention it or justify why a single question is sufficient.

Response: Thank you for the comment. We do understand your comment that the question is general. However, that is a common practice in self-reported violence measurement. And so, we had the general question of whether the individual had ever experienced any form of violence. We also went ahead to find out the specific typology of violence experienced.

• The operational definition of the dependent variable (lifetime violence) is unclear.

o Recommendation: Provide a clear operational definition of lifetime violence, specifying the types and forms included.

Response: Thank you for the comment. We have now clarified this. It reads: “Life-time experience of violence was the outcome variable. Life-time is this context means that we are not restricted to the past 12 months or 1 month preceding the survey. Rather, we were interested in whether women living with disability had ever experienced any kind of violence at any point in their life time. This was derived from the question, “Have you ever experienced any form of violence?” The response categories were yes and no. Responding yes meant that the individual had ever experienced violence in their lifetime. Additionally, the specific violence type (i.e., physical violence, sexual violence, forced/child marriage, psychosocial abuse, and spousal violence) that has ever been experienced was also covered.”

• The lifetime prevalence of violence may have recall bias.

o Recommendation: Acknowledge and discuss the potential for recall bias in the manuscript.

Response: Thank you for the comment but we had already stated in the strengths and limitations section that there is a potential for recall bias since this was based on self-reported data: “Additionally, the self-reported nature of the study and the fact that participants had to recall their experience of violence may have resulted in some recall bias”

6. Results:

• The symbol for Chi-square is incorrect.

o Recommendation: Correct the symbol for Chi-square in the results section.

Response: This has been changed. Please refer to Table 1.

• The results are presented appropriately.

Response: Thank you.

Overall Assessment:

• The manuscript addresses an important issue and uses appropriate statistical analysis. However, it requires revisions to improve clarity and detail in the methodology, keyword selection, and operational definitions.

Response: Thank you for the time spent to review the manuscript. We have addressed all of your comments.

Reviewer #2:

I greatly appreciate the authors for addressing the issue of violence against women with disabilities—a topic that is often overlooked and given low priority in our context. This study provides valuable insights into the prevalence and nature of such violence. While the paper is well-organized and adheres to journal standards, I have some observations and suggestions for improvement to make it more comprehensive.

Response: Thank you for the time spent to review our manuscript and provide valuable comments and suggestions.

Abstract

• While the abstract mentions that half of women with disabilities have experienced violence, it would be more insightful to specify which age groups are most affected and the types of violence they face, such as sexual, physical, or others, rather than focusing solely on lower odds. Additionally, there is a lack of coherence between the abstract and recommendations. The abstract emphasizes lower odds, while the recommendations highlight a high prevalence of violence and propose policies incorporating intersectionality. This inconsistency could be addressed to improve clarity and alignment.

Response: We have now indicated the age group that was most affect. It reads: “More than half (64.68%) of all participants had ever experienced violence, with girls aged 15-19 years reporting the highest prevalence (78.8%).”

Regarding the second section of your comment, we have removed the bit about intersectionality. Thank you.

Background

• Referencing within the background section is inconsistent. For example, the sixth line references a United Nations report within the text, whereas other sections adopt the Vancouver style.

Response: Thank you for drawing our attention. We have resolved this.

• Moreover, I recommend adding background information on the specific types of violence faced by women with disabilities—such as sexual, physical, or domestic violence, globally, regionally or if possible from Nepal to underline the significance of the study and its policy implications. Also, a brief discussion category of the Government of Nepal for disability, and the category that study has consider so that reader can understand why some of the disability is not consider or vice versa.

Response: We have now done this. Thank you.

Methods

• The categorization of disabilities could be improved by aligning it with the Nepalese government's guidelines, which classify disabilities into ten types. The authors should describe this framework in both the background and methods sections to provide clarity. Including a table with operational definitions for the types of violence and disabilities would also enhance the paper's comprehensiveness.

Response: We have now stated this in the background. It reads: “To bring the issue of disability into lime light, the Nepalese government in 2017 passed the Act on the Rights of Persons with Disabilities which categorizes disabilities into ten types based on the impairment (i.e., physical, vision related, hearing, deaf-blind, mental/psychological, intellectual disability, hemophilia, autism, and multiple disabilities), and also into four types based on the degree of severity [4].” However, in our study, we had no cases of autism and hemophilia. That means we had eight categories.

Additionally, the term "Brahman" should be used consistently instead of "Braman" to align with the terminology in Nepal’s national population and housing census.

Response: We have now made this correction. Please refer to Table 1 and 2.

Results

• Figures are missing from the results section, which makes it difficult to provide specific feedback. However, I suggest using the term "vocal and speech-related disability" rather than "speech/voice disability" for standard terminology.

Response: PLOS One instructs authors to attach figures as a separate file. We did that and so, you may have to contact the editor to gain access to it. However, if you manage to get access to Figure 1, you will realize that we have not used the term ‘vocal and speech-related disability’ as suggested. Thank you.

It is notable that violence is more prevalent among younger women with disabilities compared to older ones in this study which is commonly seen as well. Therefore, the discussion should explore the types of violence younger women are more susceptible to, such as sexual violence or neglect (e.g., menstrual hygiene), as this specificity would provide valuable insights.

Response: There was an error with the initial analysis. After correcting it, we realize that younger women do not have the highest prevalence of violence except in the case of sexual violence which we have now discussed. It now reads: “Our study found that as the ages of women with disability increases, the likelihood of them being exposed to violence increases. This is inconsistent with a previous study that found an inverse association between age and risk of violence victimization [29]. However, the disaggregated data shows that sexual violence was more prevalent among young girls (15-19 years) than older women. Perhaps, the high prevalence of sexual violence among young girls could be attributed to lack the autonomy, knowledge, or resources for this key population to protect themselves from sexual harassments and exploitations. at the where it found that an increase in the year of women reduces their risk of facing violence by four percent. Literature also shows that young girls living with disabilities in Nepal often lack access to sexual and reproductive health information and services. As such, they are likely to be sexually empowered, and this exacerbates their vulnerabilities to sexual violence [30].”

• While education status is not statistically significant in relation to violence, the fact that violence is slightly higher among educated women is surprising. A potential explanation could be their increased social engagement, which may expose them to stigma or bullying by peers and teachers. Discussing such non-significant in number yet intriguing findings would enrich the paper.

Response: Thank you for the suggestion. We have now discussed this counterintuitive finding. It reads: “In contrast to earlier studies that have found a significant inverse association between education and risk of violence among women with disabilities [35,36], we found no significant association. Also, women with formal education had a higher prevalence of experiencing violence. Our finding is synonymous to Puri et al.’s study [11] that found no statistically significant association between education and risk of experiencing violence among persons living with disabilities in Nepal. One possible explanation is that formal education may increase visibility and social interactions, inadvertently exposing women with disabilities to a broader spectrum of societal discrimination and violence. Moreover, systemic factors such as stigmatization or lack of enforcement of disability rights may undermine the protective role of education.”

Discussion

• The discussion section is well presented. However, I recommend incorporating an analysis of the types of violence experienced by specific age groups and discuss it more and according to provinces too. This would enable Nepal to develop targeted, evidence-based policies that are contextualized to local realities.

Response: Thank you for the comment. We have now expanded the discussion to cover all of your points. The discussion of the provincial results reads: “Compared to Province 1, women with disabilities in all other provinces were significantly less likely to experience violence, with the likelihood of experiencing violence being 96% lower among women in Sudurpaschim. To the best of our knowledge, there are no published studies directly comparing this observation. However, a study by Kakchapati et al. [30] reported that women with disabilities in Karnali and Sudurpaschim have greater access to sexual and reproductive health services. This access may indirectly contribute to reduced vulnerability to violence by exposing women to education on violence prevention, recognition of different forms of violence, and strategies to protect themselves in such circumstances.”

Conclusion

• The conclusion feels overly generic. While the results highlight factors associated with violence, the conclusion should provide more specific recommendations. For example, critical issues like denial of services should be addressed explicitly, alongside suggestions for law reform. Concluding with a focus on the significance of further research and its potential impact would make the conclusion more impactful and comprehensive.

Response: We have now revised the conclusion. It reads: “In conclusion, the experience of violence varies significantly by disability type. There is a need for comprehensive, disability-inclusive policies that address the specific risks identified. Policy interventions in Nepal must focus on improving access to services for women with vocal and speech-related disabilities, ensuring that healthcare and support services are inclusive of alternative communication methods. Additionally, tailored support for divorced or separated women and age-appropriate interventions for young girls with disabilities are crucial in mitigating their increased vulnerability to violence.”

---

## [Editor Report · Decision Letter 1]

Dear Dr. Okyere,

Thank you for submitting your manuscript to PLOS ONE. After careful consideration, we feel that it has merit but does not fully meet PLOS ONE’s publication criteria as it currently stands. Therefore, we invite you to submit a revised version of the manuscript that addresses the points raised during the review process.

We look forward to receiving your revised manuscript.

Kind regards,

Shalik Ram Dhital, PhD

Academic Editor

PLOS ONE

Reviewers' comments: Please address each reviewer's comments as provided. 

---

## [Decision Letter · Decision Letter 2]

Dear Dr. Okyere,

Thank you for submitting your manuscript to PLOS ONE. After careful consideration, we feel that it has merit but does not fully meet PLOS ONE’s publication criteria as it currently stands. Therefore, we invite you to submit a revised version of the manuscript that addresses the points raised during the review process.

Please response all the comments and feedbacks from the reviewers. 

We look forward to receiving your revised manuscript.

Kind regards,

Kshitij Karki, MPH, MA

Academic Editor

PLOS ONE

Reviewers' comments:

Reviewer's Responses to Questions

**Comments to the Author**

Reviewer #3: (No Response)

Reviewer #4: All comments have been addressed

2. Is the manuscript technically sound, and do the data support the conclusions?

Reviewer #3: Yes

Reviewer #4: Partly

3. Has the statistical analysis been performed appropriately and rigorously?

Reviewer #3: Yes

Reviewer #4: No

4. Have the authors made all data underlying the findings in their manuscript fully available?

Reviewer #3: Yes

Reviewer #4: No

5. Is the manuscript presented in an intelligible fashion and written in standard English?

Reviewer #3: Yes

Reviewer #4: Yes

Reviewer #3: The manuscript broadly satisfies the publication criteria. There are some remaining issues (misinterpretation, missing information) that would significantly strengthen the publication.

Reviewer #4: Simkhada et al. Life-time experience of violence among women and girls living with disability in Nepal: A cross-sectional study

This is a straightforward study reporting life-time prevalence of violence among women and girls living with disabilities. However, there is one major issue with the paper, that needs to be rectified.

The article lacks a table on distribution by types of disabilities. The description on how disability was identified is poorly explained. Authors should clearly describe if some method was used to confirm disability, eg disability pension books, or disability card (if available).

The authors have mentioned that they had excluded severe intellectual disability, but elsewhere the paper states that women and girls with intellectual disabilities were at a higher risk of sexual violence.

The authors should note that without clarification on how disability was measured, and the distribution of disability by type, the section on “type of disability and violence experienced” is not meaningful. The authors should incorporate these statistics in appropriate sections of the manuscript.

**Do you want your identity to be public for this peer review?** For information about this choice, including consent withdrawal, please see our Privacy Policy

Reviewer #3: No

Reviewer #4: **Yes: ** Prof Anita Kar

---

## [Author Response · Author response to Decision Letter 3]

15 May 2025

Referee report: Life-time experience of violence among women and girls living with disability in Nepal: A cross-sectional study

The authors conduct an important descriptive study of the life-time experience of violence among women and girls living with disability in Nepal, shedding light on the prevalence rates for different demographics and regions in a national survey sample and on the predictive factors associated with experiencing violence among this vulnerable population. Overall, the study is presented in an intelligible fashion and analyses are described well. My review has highlighted a few issues that – while major, as in, critical to meet publication standards – the authors should be able to address easily.

Response: Thank you for the kind words, and for the time spent to review the revised manuscript. We appreciate the constructive feedbacks.

These major issues include, in order of appearance:

• The result “Participants who had not witnessed violence against other women exhibited significantly lower odds of experiencing violence” is at odds with the AOR reported in Table 2 – these women appear to have higher odds of experiencing violence. This result is mentioned in the Abstract as well as Results sections.

Response: Thank you for drawing our attention. This error has been rectified. It reads: “Participants who had not witnessed violence against other women exhibited significantly higher odds of experiencing violence”.

• It would be helpful for context to understand a bit more what being “part of a larger project in Nepal to reduce violence against women with disabilities” means for the survey. Does this mean that the survey was longer than the questions presented here?

What does “project interventions” refer to, which are mentioned as a reason for an estimated 5% reduction in violence against women with disability? Does this mean that this survey comes after (experimental/government/NGO) interventions were run to reduce violence? Or is this survey the “baseline” for the larger project?

Response: Thank you for the comment. This study is part of a project that is investigating the wellbeing of women and girls living in Nepal. It covers health and social issues including violence experience. We have now indicated this: “It must be noted that this study is a baseline survey of a larger project in Nepal exploring multiple social and health dimensions of women and girls, including those living with disabilities”.

• I would like to know/see some explanation in the manuscript to whether the question about life-time experience of any form of violence was accompanied by any sort of explanation about what constitutes violence. If not, I think it is a limitation that some

girls and women may not have recognized that they were subject to violence (e.g., denial of services, psychological), and so may have answered ‘no’ to the general question and were subsequently not asked for any further detail of what specific violence type they may have experienced. In that case, the prevalence could potentially be higher and also the relative occurrence of different forms of violence could be skewed. If this is the case, this should be acknowledged in the limitations.

Response: Thank you for this insightful comment. The question did not come with any other explanations. As such, we have acknowledged this in the limitations section. It reads: “Another limitation of this study lies in how life-time experience of violence was measured. There was no prompting in the questioning. As such, it is possible that some participants may not recognize some actions (e.g., denial of service) as violence. Therefore, it is possible that the prevalence may be underestimated in this study.”

• Table 1 and associated text:

o I would suggest being careful interpreting significant differences where groups are very small, i.e., the Kirat group with just over 20 observations in total.

Response: Thank you for the caution. However, as you may have realized, the distribution was based on row % rather than column %. That approach addresses the dangers that can potentially arise when interpreting the significant differences as you pointed out.

o The numbers in ‘Age at first marriage’ are strange; should the total for whom this is available not be equal to 550 (Currently married)+35 (Divorced)+57

(Widowed), so, excluding the 652 who were ‘never married’? If data is available for the full sample, it is unclear what the variable actually means.

Response: Thank you for drawing our attention. This correction has been made. Please see the table here.

Age at first marriage 0.002

Never married 652 (50.39) 206 (31.60)

Before 20 years 559 (46.29) 228 (38.06)

At 20 year or older 43 (3.32) 23 (53.49)

o Minor comment: There must be a typo in the percentage of “No” for “Knowledge on policies and laws that support victims of violence” – it cannot be 100%.

Response: Yes. That is quite observant of you. We have addressed this. Thank you: “1025 (79.21)”

• Table 2 and associated text:

o I would make clear that this was a multi-select question that was asked of the 457 women and girls who had experienced any violence. That would then help to understand why the percentages add up to more than 100%, and what the total N is.

Response: Thank you. We have now added a footnote this this: “NB: This was a multi-select question answered by 457 participants who had every experienced any violence”

• As above, I would refrain from interpreting types of violence with very low sample, i.e., sexual violence (n=10) and forced/child marriage (n=12) (At least add the caveat that percentages in small samples are not reliable, as one single observation can change the percentage dramatically.)

Response: Thank you for the suggestion. In the revised manuscript, we have refrained from interpreting the results with a low sample.

• Type of disability and violence experienced: I think this is an interesting and important section, and it would deserve a bit more insight. As a reader, I am wondering what the distribution is of different types of disability in the sample. How common is each type of disability? And how many girls/women have more than one disability type? My

suggestion would be to add these variables to Table 1.

Response: We have now included this in Table 1 as suggested: “Specifically, 42.08% of women with a visual disability reported experiencing violence, compared to 33.62% of women without a visual disability. Similarly, 37.65% of women with a physical disability experienced violence, compared to 33.33% among those without. However, for self-care disability, the prevalence of violence was similar between women with the disability (33.92%) and those without (35.95%). In contrast, lower prevalence of violence was observed among women with hearing (23.66% vs. 36.22%), cognitive (24.46% vs. 36.62%), and communication disabilities (19.05% vs. 37.40%) compared to their counterparts without these disabilities.”

• Table 3 and associated text:

o Please explain the backward stepwise approach that led to exclusion of certain variables. This is not mentioned anywhere in the methods or table notes.

Response: We have now included a description of this in the methods section: “In the multivariable analysis, we employed a backward stepwise logistic regression approach to build the final adjusted model. Initially, all variables with theoretical relevance and those significant at p<0.20 in the bivariate analysis were included in the full model. Variables were then sequentially removed based on the likelihood ratio test (LRT) and Akaike Information Criterion (AIC) values, ensuring that removal did not meaningfully alter the effect estimates of key predictors or degrade the model's fit. Variables with p-values above 0.05 that did not contribute significantly to model fit and lacked theoretical justification were excluded. Specifically, religion, education, age at first marriage, disability allowance receipt, and knowledge on policies and laws supporting victims were excluded in the final model due to lack of statistical significance and negligible impact on the effect estimates of retained variables”

o As a robustness test, I would suggest using a different province to Koshi Province as reference category. The reason why is that you focus a lot on Koshi Province (rightly so because it clearly has the highest prevalence, and the highest

adjusted odds), but Table 1 shows that Madhesh and Bagmati Province also have high prevalence rates (around 50%), much higher than the remaining provinces and much higher than the sample average. The fact that the odds for provinces do not change a lot when adjusted for other variables indicates to me that you might be missing some important recommendations for other provinces, that also have quite high prevalence, but show up as ‘low odds’ simply because they are compared to Koshi Province. I would choose Karnali Province (= the Province with the prevalence rate most closely aligned with the whole sample average) as reference category in the logistic regressions.

Response: Thank you for the different perspective. We have now done this: “Women living in the Koshi province [AOR=4.04; 95%CI: 2.54-6.42], Madhesh province [AOR=2.16; 95%CI: 1.15-4.08] and Bagmati province [AOR=2.21; 95%CI: 1.41-3.46] reported significantly higher odds of experiencing violence compared to those in Karnali.”

• When you discuss the association of experience of violence with age, your conclusion that this is inconsistent with reference [29] is misguided. That paper examines experience of violence in the last 5 years, not lifetime. In your setup, life-time exposure to violence increases necessarily with age, by design (= theoretically, it can never go down, only up or stay the same), and it says nothing about which age groups are more likely to being exposed to violence. Please avoid making such statements. Similarly, when discussing a higher prevalence of sexual violence among young girls, ‘lack of autonomy’ would have applied to older women equally when they were young. So, the fact that they haven’t reported sexual violence indicates either recall bias or that “the times are changing”, perhaps that sexual violence is much more of an issue nowadays than it was 30-50 years ago.

Response: We have revised this section accordingly: “The difference between our findings and that of Brownridge [29] stems from the point that unlike the cited study that focused on experience of violence in the last five years, our study focused on life-time experience. And so, our finding reflects the cumulative nature of lifetime exposure, where the probability of ever experiencing violence increases with age due to a longer period of risk exposure. Therefore, this association should be interpreted cautiously, as it does not imply that older women are currently at higher risk of violence victimization compared to younger women, but rather reflects accumulated experiences over the life course.”

Minor issues include:

- A few typos throughout the manuscript. Please review carefully. There is also an incomplete sentence in the fourth paragraph of the Discussion (“at the where it found that….”).

Response: We have now addressed the typos in the manuscript. Thank you.

- Reference 18 (pilot studies) lists an incorrect year of publication (2015 instead of 2005)

Response: We have now addressed this: “van Teijlingen E, Hundley V. Pilot studies in family planning and reproductive health care. BMJ Sexual & Reproductive Health. 2005 Jul 1;31(3):219.”

Reviewer #4: Simkhada et al. Life-time experience of violence among women and girls living with disability in Nepal: A cross-sectional study

This is a straightforward study reporting life-time prevalence of violence among women and girls living with disabilities. However, there is one major issue with the paper, that needs to be rectified.

Response: Thank you for the time spent to review the revised manuscript, and for the constructive feedbacks.

The article lacks a table on distribution by types of disabilities. The description on how disability was identified is poorly explained. Authors should clearly describe if some method was used to confirm disability, eg disability pension books, or disability card (if available).

Response: Thank you. We have now included a description of how disability was measured: “Disabilities were assessed across six functional domains: seeing, hearing, walking/climbing steps, remembering/concentrating, self-care, and communication. For each domain, participants were asked about the level of difficulty experienced, with responses categorized as: no difficulty, some difficulty, a lot of difficulty, or cannot do at all. Following the Washington Group recommendations, a binary variable was created for each domain, coded as 1 if the respondent reported “a lot of difficulty” or “cannot do at all,” and 0 otherwise. Subsequently, a composite variable was generated to identify individuals with two or more disabilities by summing the six domain-specific binary variables and recoding the total as 1 if the individual had disabilities in two or more domains, and 0 otherwise.”

The authors have mentioned that they had excluded severe intellectual disability, but elsewhere the paper states that women and girls with intellectual disabilities were at a higher risk of sexual violence.

Response: Thank you for the comment. However, there is no contradiction in the manuscript. As stated, women and girls with severe intellectual disabilities were excluded due to the lack of trained mental health interviewers capable of supporting their specialized needs during data collection. The analyses presented in the paper refer to women and girls with mild to moderate cognitive disabilities (as measured by self-reported difficulty in remembering or concentrating). Therefore, references to an increased risk of violence pertain to this group, not to those with severe intellectual impairments who were excluded from the study.

The authors should note that without clarification on how disability was measured, and the distribution of disability by type, the section on “type of disability and violence experienced” is not meaningful. The authors should incorporate these statistics in appropriate sections of the manuscript.

Response: Thank you for the comment. This detail has been included as suggested.

---

## [Decision Letter · Decision Letter 3]

Life-time experience of violence among women and girls living with disability in Nepal: A cross-sectional study

PONE-D-24-24275R3

Dear Dr. Joshua Okyere,

We’re pleased to inform you that your manuscript has been judged scientifically suitable for publication and will be formally accepted for publication once it meets all outstanding technical requirements.

Kind regards,

Kshitij Karki, MPH, MA

Academic Editor

PLOS ONE

Additional Editor Comments (optional):

Please address the minor comments from reviewer.

Reviewers' comments:

Reviewer's Responses to Questions

**Comments to the Author**

Reviewer #3: (No Response)

Reviewer #4: All comments have been addressed

2. Is the manuscript technically sound, and do the data support the conclusions?

Reviewer #3: Yes

Reviewer #4: Yes

3. Has the statistical analysis been performed appropriately and rigorously?

Reviewer #3: Yes

Reviewer #4: Yes

4. Have the authors made all data underlying the findings in their manuscript fully available?

Reviewer #3: Yes

Reviewer #4: Yes

5. Is the manuscript presented in an intelligible fashion and written in standard English?

Reviewer #3: No

Reviewer #4: Yes

Reviewer #3: Thank you for the revision, all comments have been addressed. I only noticed a minor error where this revised sentence in the manuscript is lacking the comparator percentages:

"Specifically, 42.08% of women with a visual disability reported experiencing violence, compared to those without a visual disability. Similarly, 37.65% of women with a physical disability experienced violence, compared to those without such disability. "

The Author Response has these comparator percentages, so it's just a matter of copy-and-pasting.

Reviewer #4: The authors have satisfactorily answered all comments. There are no additional revisions needed from my end.

**Do you want your identity to be public for this peer review?** For information about this choice, including consent withdrawal, please see our Privacy Policy

Reviewer #3: No

Reviewer #4: **Yes: ** Anita Kar

---

## [Editor Report · Acceptance letter]

PONE-D-24-24275R3

PLOS ONE

Dear Dr. Okyere,

I'm pleased to inform you that your manuscript has been deemed suitable for publication in PLOS ONE. Congratulations! Your manuscript is now being handed over to our production team.

Kind regards,

on behalf of

Dr. Kshitij Karki

Academic Editor

PLOS ONE